# Pulse wave velocity is a new predictor of acute kidney injury development after off-pump coronary artery bypass grafting

**Jae-Sung Choi** [ID] **, Se Jin Oh, Yong Won Sung, Hyeon Jong Moon, Jeong Sang Lee** [ID] *

Department of Thoracic and Cardiovascular Surgery, Boramae Medical Center, Seoul National University College of Medicine, Seoul, South Korea

* jeongsl@snu.ac.kr

**Data Availability Statement:** All relevant data are within the paper and its Supporting Information files.

**Funding:** The authors received no specific funding for this work.

## Abstract

### Background

Brachial-ankle pulse wave velocity (baPWV) is the simple, non-invasive, gold-standard method for assessing arterial stiffness. However, baPWV has been shown to be associated with renal dyfunction, with a few reports demonstrating an association between baPWV and postoperative acute kidney injury (AKI) among surgical patients.

### Methods

We retrospectively analyzed preoperative baPWV data that were prospectively collected from 164 patients who underwent off-pump coronary artery bypass grafting (CABG) between April 2013 and July 2019 (mean age: 66.2 ± 10.3 years, 29.3% females). Primarily, baPWV was investigated as an independent predictor of postoperative AKI development; secondarily, the patients were divided into high and low PWV groups according to the optimal baPWV cut-off value. Postoperative complications, mortality, and mid-term survival were compared between the two groups.

### Results

AKI developed in 30 patients (18.3%). Univariate analysis showed that AKI was significantly associated with baPWV (20.2±7.3 vs. 16.2±2.8 m/s, $p < 0.001$), age, preoperative serum creatinine, and EuroSCORE. Multivariable logistic regression analysis revealed baPWV as independently associated with postoperative AKI even after adjustment for preoperative creatinine, old age (> 75 years), hypertension, diabetes under insulin therapy, and EuroSCORE. Moreover, area under the curve (AUC) analysis indicated that PWV can predict AKI better than preoperative creatinine levels (AUC, 0.781 [95% confidence interval, 0.688–0.874] vs. 0.680 [0.568–0.792]). The group-dividing baPWV cut-off value for AKI was 19 m/s. There were no 30-day mortality. The in-hospital mortality rates in the high and the low PWV groups were 2.2% (n = 1) and 0.8% (n = 1), respectively ($p = 0.484$). Midterm survival rates were not different between the two groups, but the rate of composite neurologic

**Competing interests:** The authors have declared that no competing interests exist.

complication composed of stroke and delirium, was higher, and rate of mechanical ventilatory support was longer, in the high PWV group.

## Conclusion

Brachial-ankle pulse wave velocity was an independent predictor of postoperative AKI following off-pump CABG, and high baPWVs may affect the composite neurologic outcome and the duration of mechanical ventilatory support.

## Introduction

Postoperative acute kidney injury (AKI) develops at a considerable rate, in up to 30% of patients [1]. Despite advances in the management of AKI, it contributes to increased mortality rates and poor patient outcomes after cardiac surgery [2]. This serious complication reflects the intersection of renal ischemia, reperfusion injury, athero-embolism, leukocyte recruitment from systemic inflammation, and oxidative stress, which are caused by multiple perioperative risk factors. Some of these factors are modifiable by early detection [3], whereas others are not, such as patients' age and disease status—for example, if they have hypertension, diabetes, or chronic kidney disease. Predicting and preventing AKI is essential to improving surgical outcomes, considering that outcomes may worsen when even mild AKI is disregarded.

Most studies investigating the use of pulse wave velocity (PWV) to estimate arterial stiffness and the association between PWV measurement and renal function have examined only non-surgical patients [4]. We and others performed the first prospective studies to validate PWV as a reliable marker for predicting renal insufficiency after cardiac surgery [3,5].

This study aimed to evaluate preoperatively measured brachial-ankle PWV (baPWV) as a potential independent predictor of AKI following surgery. Secondarily, the associations between baPWV and postoperative complications, mortality, and mid-term survival were also investigated. We only analyzed patients who underwent off-pump coronary artery bypass grafting (CABG) to avoid the confounding effects of cardiopulmonary bypass.

## Methods

### Patient selection

The selection of the study sample is shown in Fig 1. To avoid major bias from the influence of cardiopulmonary bypass on postoperative AKI, on-pump CABG cases were excluded. We retrospectively reviewed the electronic medical records of 243 patients who underwent isolated off-pump CABG between April 2013 and July 2019. Our exclusion criteria were as follows: 1) when PWV measurements were not taken, 2) insertion of aorto-iliac or renal stents (grafts), 3) when patients had oliguria or already started dialysis, 4) combination with any other cardiac procedure, and 5) when patients had uncontrolled, severe hypertension (blood pressure >160/100 mmHg). Based on these predetermined criteria, 79 patients were excluded, and the remaining 164 patients were included in this study. The median follow-up duration was 39.2 months (range, 1.6–78.0 months). There was no loss to follow-up among the included patients.

The study protocol was approved by the Seoul Metropolitan Government—Seoul National University Hospital's institutional review board, and the requirement for informed consent

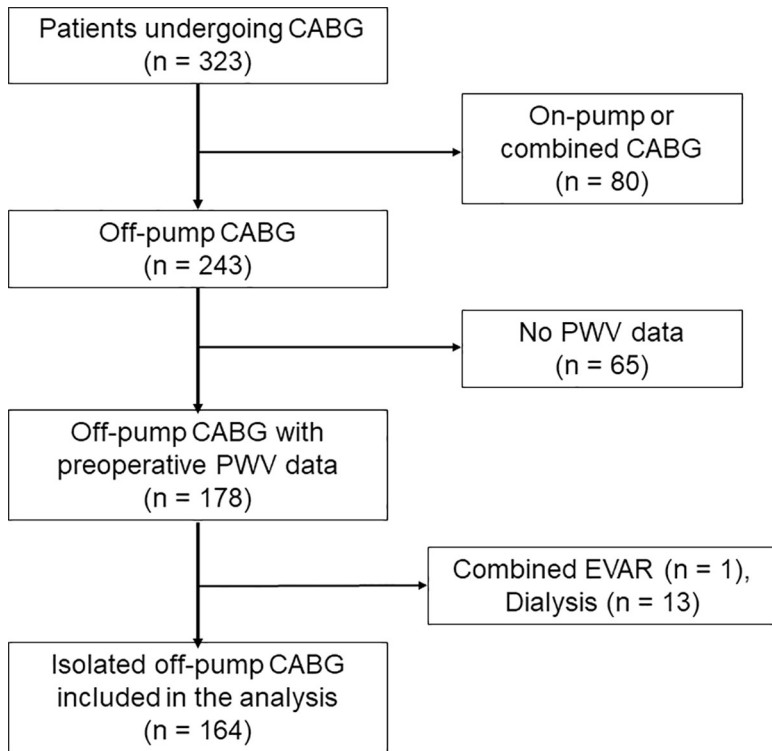

**Fig 1. Patient selection.** CABG, coronary artery bypass grafting; EVAR, endovascular abdominal aortic repair; PWV, pulse wave velocity.

was waived because this retrospective review of medical records could not adversely affect the rights or welfare of the subjects.

## Measurement of baPWV

As described previously by our group [5], baPWV measurements were preoperatively performed using a volume-plethysmographic apparatus (VP-1000, Colin Co. Ltd.; Komaki, Japan) in accordance with the manufacturer's instructions. Cuffs were wrapped on both brachialis muscles and ankles of the patients, and recordings of pulse volume waveform, blood pressure, phonogram, and heart rate were performed simultaneously. In other words, the PWV value can be obtained by simple wrapping cuffs around the four extremities. To calculate the baPWV, the virtual length of the arterial path between the heart and the brachial artery (Lhb) and the length of the path between the heart and the posterior tibial artery (Lha) should be derived from the subject's height using the appropriate formulas. Then, the time difference between the beginning points of systolic increase in brachial and ankle pressure waves (Tba) should be measured. Finally, baPWV was calculated by the path length difference divided by the time difference (baPWV = (Lha–Lhb)/Tba). Relative to clinically normal limits, higher PWV values indicated that arterial stiffness was more severe. The mean values between the left and right baPWVs were used for analysis in this study.

## Surgical techniques

Our main strategy during the study period was to use the saphenous vein (SV) as part of a composite graft based on an *in situ* left internal thoracic artery (LITA). To avoid size-mismatches between harvested SVs and target coronary arteries, the SVs were harvested from

lower legs rather than from the thigh. The SV was connected to the in situ LITA in a Y- or an I-shape and then was anastomosed sequentially to the target points except for the left anterior descending artery, which was exclusively revascularized by the LITA. The off-pump CABG was basically performed with this no-touch aorta technique, but sometimes aortic manipulation, including partial clamping or using a Heatstring III proximal seal system (Maquet holding B.V. & Co., Rastatt, Germany), was necessary when an additional inflow source was needed—for example, when flow competition was expected or recognized during a transit-time flow measurement after completing sequential anastomosis. For relatively young patients (less than 60 years of age), arterial grafts like the right internal thoracic artery or radial artery, were favored over SV grafts.

## Renal function and postoperative complications

Renal function was assessed by serum creatinine and estimated glomerular filtration rate (eGFR) using the Modification of Diet in Renal Disease (MDRD) equation. Postoperative GFR was defined as the lowest eGFR within 7 postoperative days (PODs). According to the 2012 Kidney Disease: Improving Globall Outcomes (KDIGO) Foundation consensus statement [6], AKI was defined at a stage of 1 or higher, by any of the following benchmarks: urine output <0.5 mL/kg/hr for 6 hours or longer; elevation of serum creatinine within 2 PODs >0.3 mg/dL; and a serum creatinine increase >1.5 times relative to the baseline value within 7 PODs.

Postoperative pneumonia was defined as a lower respiratory tract infection with accompanying consolidation detected on chest x-ray. Delirium was diagnosed according to the criteria of the fifth edition of the Diagnostic and Statistical Manual of the American Psychiatric Association (DSM-5) and confirmed by consultation with a neuropsychiatrist. Perioperative myocardial infarction was defined based upon an elevation of biomarkers (either creatine kinase (CK-MB) concentration >40 ng/mL or peak troponin I levels >15 ng/mL at 12 hours after operation) and the presence of new pathological Q waves or left bundle branch block. Late mortality was defined as any-cause mortality after POD 30.

## Statistical analysis

Descriptive statistics for continuous variables are expressed as mean ± standard deviation, or median (range) if the data were not found to be normally distributed using the Kolmogorov–Smirnov test. Categorical variables are expressed as numbers with proportions (%). To examine the associations of baseline characteristics, including baPWV, with AKI, univariate analysis was performed using either the independent samples t-test or the Mann–Whitney $U$-test for comparison of continuous variables, or either Pearson's chi-square test or Fisher's exact test for categorical variables. Only the variables that were found to have significant associations with AKI in the univariate analysis were selected and then analyzed together with baPWV in a multivariable logistic regression analysis to determine whether baPWV was an independent predictor of AKI. At this time, only serum creatinine concentration was selected as an explanatory variable instead of eGFR values, to avoid problems of multi-co-linearity. The optimal baPWV cut-off value was determined using the receiver operating characteristic (ROC) curve and, using this, the subjects were divided into high and low PWV groups. Postoperative complications were compared between the two groups with typical univariate analysis methods. Unadjusted patient survival was estimated using Kaplan–Meier methods and between-group comparisons were performed using the log-rank test. A $p$-value <0.05 was considered statistically significant. Analyses were performed using SPSS Statistics for Windows, version 20.0 (IBM Corp., Armonk, NY, USA) and R Studio, version 1.2.5001 (R Foundation for Statistical Computing, Vienna, Austria), with various R packages.

## Results

### Association of baseline characteristics with postoperative AKI

Of 164 patients, thirty patients developed AKI (18.3%). Table 1 shows the differences in baseline characteristics between the patients with and without postoperative AKI. Patients with AKI were older (72.8 ± 5.0 years vs. 64.7 ± 10.7 years, p < 0.001) and had higher preoperative serum creatinine levels (1.1 [0.6–6.2] mg/dL vs. 0.9 [0.5–10.9] mg/dL, $p = 0.002$) than patients without AKI. Additionally, among patients with AKI, advanced chronic kidney disease (stage ≥4) was more common (33.3%, n = 10 vs. 7.5%, n = 10; $p = 0.002$), and the median EuroSCORE II was higher (2.3 [0.9–14.1] vs. 1.4 [0.5–21.4], $p < 0.001$). The mean baPWV was also higher (20.2 ± 7.3 m/s vs. 16.2 ± 2.8 m/s, $p < 0.001$) among AKI patients. The factors that were not statistically significant but showed some trends toward significance were hypertension ($p = 0.067$), and diabetes under insulin therapy ($p = 0.076$). However, there were no differences

**Table 1. Baseline characteristics of patients with and without postoperative AKI.**

|  | No AKI (n = 134) | AKI (n = 30) | *p*-value |
|---|---|---|---|
| Female, n (%) | 37 (28) | 11 (37) | 0.445 |
| Age, years | 64.7±10.7 | 72.8±5.0 | < 0.001 |
| Age > 75 years, n (%) | 21 (16) | 11 (37) | 0.018 |
| Body mass index, kg/m$^2$ | 24.1±3.2 | 24.3±3.5 | 0.805 |
| Obesity, n (%) | 59 (44) | 14 (47) | 0.953 |
| Hypertension, n (%) | 91 (68) | 26 (87) | 0.067 |
| Diabetes, n (%) | 67 (50) | 20 (67) | 0.147 |
| *under insulin therapy, n (%)* | 16 (12) | 8 (27) | 0.076 |
| Dyslipidemia, n (%) | 42 (31) | 12 (40) | 0.486 |
| History of cerebrovascular accidents, n (%) | 20 (15) | 6 (20) | 0.681 |
| Peripheral arteriopathy, n (%) | 35 (26) | 10 (33) | 0.566 |
| Chronic obstructive pulmonary disease, n (%) | 8 (6) | 2 (7) | 1.000 |
| Unstable angina, n (%) | 86 (64) | 18 (60) | 0.668 |
| Acute MI, n (%) | 18 (13) | 6 (20) | 0.526 |
| Recent MI, n (%) | 11 (8) | 4 (13) | 0.596 |
| Old MI, n (%) | 6 (5) | 2 (7) | 0.973 |
| Single-vessel disease, n (%) | 6 (5) | 1 (3) | 1.000 |
| Double-vessel disease, n (%) | 19 (14) | 7 (23) | 0.215 |
| Triple-vessel disease, n (%) | 109 (81) | 22 (73) | 0.323 |
| Left main disease, n (%) | 45 (34) | 11 (37) | 0.747 |
| History of coronary intervention, n (%) | 17 (13) | 5 (17) | 0.778 |
| Atrial fibrillation, n (%) | 1 (1) | 1 (3) | 0.805 |
| Left ventricle ejection fraction (%) | 57.2±13.0 | 57.7±11.9 | 0.842 |
| Creatinine, mg/dL | 0.9 (0.5–10.9) | 1.1 (0.6–6.2) | 0.002 |
| MDRD-GFR, mL/min/1.73 m$^2$ | 86.9 (5.1–155.2) | 65.6 (6.9–106.1) | < 0.001 |
| Chronic kidney disease stage ≥ 4 | 10 (7.5) | 10 (33.3) | 0.002 |
| Mean baPWV, m/s | 16.2±2.8 | 20.2±7.3 | < 0.001 |
| *right baPWV, m/s* | 16.3±3.6 | 20.6±5.5 | < 0.001 |
| *left baPWV, m/s* | 16.1±3.4 | 22.1±9.5 | 0.002 |
| EuroSCORE II | 1.4 (0.5–21.4) | 2.3 (0.9–14.1) | < 0.001 |

AKI, acute kidney injury; baPWV, brachial-ankle pulse wave velocity; GFR, glomerular filtration rate; MDRD, Modification of Diet in Renal Disease; MI, myocardial infarction.

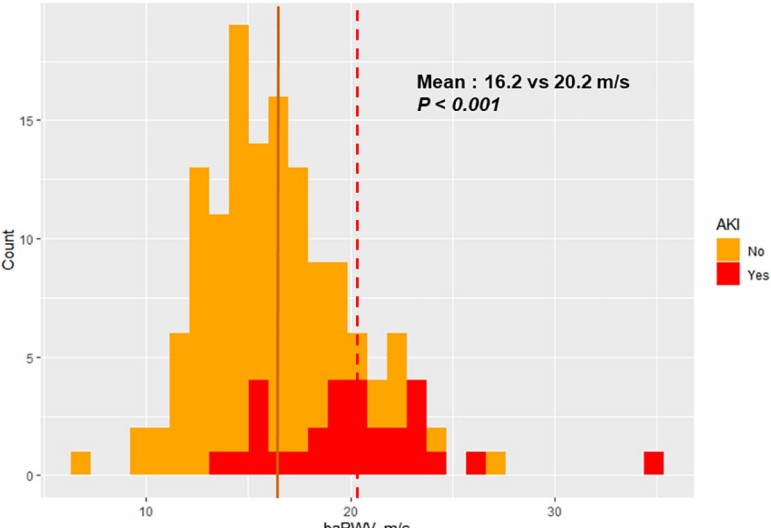

**Fig 2. Comparison of distributions of baPWV data between patients with and without acute kidney injury.** Solid line = mean baPWV of patients without AKI, dotted line = mean baPWV of patients with AKI. AKI, acute kidney injury; baPWV, brachial-ankle pulse wave velocity.

in the frequencies of unstable angina, myocardial infarction, and extent of coronary vessel involvement between the two groups.

The compared distributions of baPWV data between the patients with and without AKI are depicted in Fig 2. The negative correlation between baPWV and postoperative eGFR is presented in Fig 3, where analysis was limited to the patients whose preoperative eGFRs were ≥50 mL/min/1.73m$^2$ (N = 138), to remove the influence of low preoperative eGFR outliers on this correlation.

There were no statistical differences between the patients with and without postoperative AKI in terms of the surgical techniques, the types of grafts used, and the target coronary arteries (Table 2).

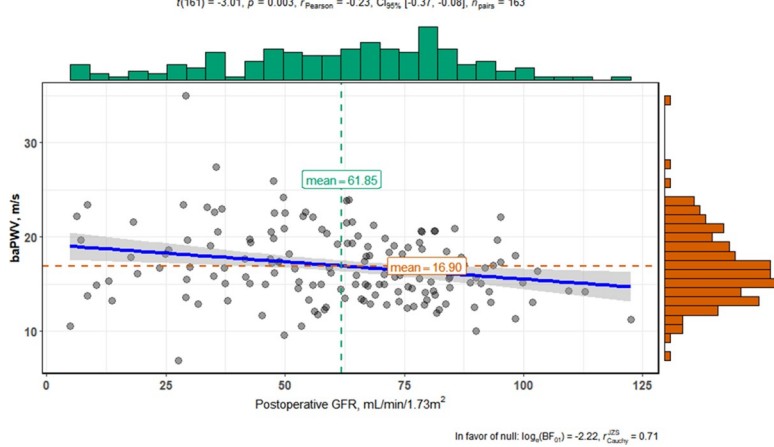

**Fig 3. Correlation between brachial-ankle pulse wave velocity and postoperative glomerular filtration rate.** Subjects were limited to patients with preoperative eGFR ≥ 50 mL/min/1.73m$^2$ (n = 138). γ = Pearson's coefficient of correlation. baPWV, brachial-ankle pulse wave velocity; CI, confidence interval; GFR, glomerular filtration rate.

**Table 2. Comparison of surgical techniques, used grafts, and target vessels.**

| | No AKI (n = 134) | AKI (n = 30) | *p*-value |
|---|---|---|---|
| Urgency or emergency, n (%) | 5 (4) | 1 (3) | 1.000 |
| Minimally invasive bypass, n (%) | 2 (1) | 0 (0) | 1.000 |
| No touch aorta technique, n (%) | 121 (90) | 26 (87) | 0.555 |
| Use of grafts | | | |
| *left internal thoracic artery, n (%)* | 125 (93) | 27 (90) | 0.532 |
| *right internal thoracic artery, n (%)* | 37 (28) | 7 (23) | 0.633 |
| *radial artery, n (%)* | 7 (5) | 0 (0) | 0.352 |
| *saphenous vein, n (%)* | 112 (84) | 28 (93) | 0.172 |
| Target coronary artery | | | |
| *left anterior descending* | 130 (97) | 29 (97) | 0.920 |
| *diagonal branch* | 42 (31) | 9 (30) | 0.886 |
| *ramus intermedius* | 9 (7) | 4 (13) | 0.260 |
| *left circumflex/obtuse marginal* | 93 (69) | 20 (67) | 0.770 |
| *right coronary* | 94 (70) | 21 (70) | 0.987 |
| # of distal anastomosis | 2.9±1.0 | 3.0±0.9 | 0.814 |

AKI, acute kidney injury.

## Independent predictors of postoperative AKI

In the multivariable logistic regression analysis, baPWV remained a statistically significant predictor of AKI (Exp [ß] = 1.34; 95% confidence interval [CI], 1.17–1.58; *p* < 0.001) even after adjusting for preoperative creatinine level, old age (> 75 years), hypertension, diabetes under insulin therapy, and EuroSCORE II (Table 3). This statistical independence of baPWV as a predictor of AKI was also maintained when preoperative eGFR was entered into the multivariable regression model instead of serum creatinine. Although preoperative serum creatinine concentrations maintained significance in conjunction with baPWV after the multivariable analysis, when comparing area under the curve (AUC) values, baPWV was noted to predict AKI better than creatinine level (AUC, 0.781 [95% CI, 0.688–0.874] vs. 0.680 [95% CI, 0.568–0.792]) (Fig 4). Moreover, using preoperative baPWV and serum creatinine cut-off values of 18.2 m/s and 1.0 mg/dL, respectively, from the respective ROC curves, the positive and negative predictive values for baPWV are 43.5% and 91.5%, respectively, compared with 32.7% and 88.4%, for creatinine. Therefore, baPWV is likely to be more useful in terms of predicting postoperative AKI.

**Table 3. Multivariable risk factor analysis for postoperative acute kidney injury.**

| Variables | ß | SE | Exp (ß) | 95% CI | *p*-value |
|---|---|---|---|---|---|
| Age > 75 years | 1.009 | 0.578 | 2.741 | 0.867–8.543 | 0.081 |
| Hypertension | 0.574 | 0.650 | 1.775 | 0.532–7.203 | 0.377 |
| Diabetes on insulin | 0.174 | 0.623 | 1.190 | 0.336–3.916 | 0.780 |
| Creatinine level | 0.411 | 0.148 | 1.508 | 1.122–2.061 | 0.006 |
| baPWV | 0.294 | 0.077 | 1.341 | 1.167–1.579 | < 0.001 |
| EuroSCORE II | 0.021 | 0.089 | 1.022 | 0.847–1.238 | 0.812 |

baPWV, brachial-ankle pulse wave velocity; CI, confidence interval; Multivariable logistic regression analysis; SE, standard error.

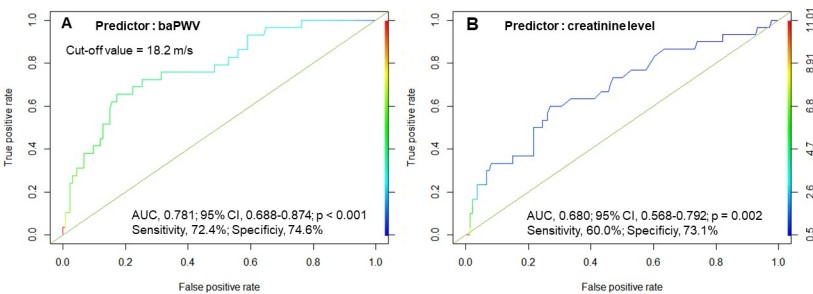

**Fig 4. Comparison of receiver operating characteristic curves in the selection of predictors of postoperative acute kidney injury. AUC, area under the curve; CI, confidence interval; PWV, pulse wave velocity.**

## Impact of high baPWV on postoperative outcomes

The optimal baPWV cut-off value for predicting AKI was determined as 18.2 m/s from the ROC analysis. Thus, 19 m/s was stated as the dividing point between designations of "high" and "low" for PWV values for each investigative group. When comparing postoperative complications, as demonstrated in Table 4, AKI developed more frequently among patients in the high PWV group (41.3%, n = 19 vs. 9.3%, n = 11, $p < 0.001$) despite the fact that preoperative creatinine levels of the high PWV group was not statistically different from those of the low PWV group, and preoperative median eGFR value of the high PWV group was 75.7 mL/min/1.73 $m^2$ which was not bad. There was no patient who required renal replacement therapy in patients with postoperative AKI though. For further details about comparison of baseline characteristics between the high and the low baPWV groups, please see supplementary material (S1 Table). Similarly, the composite incidence of stroke and delirium (composite neurologic

**Table 4. Inter-group comparison of postoperative complications and motality.**

|  | n (%) | Low baPWV | High baPWV | *p*-value |
|---|---|---|---|---|
|  | (N = 164) | (n = 118) | (n = 46) |  |
| Acute kidney injury | 30 (18.3) | 11 (9.3) | 19 (41.3) | < 0.001 |
| *KDIGO stage 1* | 26 (15.9) | 8 (6.8) | 18 (39.1) | < 0.001 |
| *KDIGO stage 2* | 4 (4.3) | 2 (1.7) | 2 (4.3) | 0.313 |
| *KDIGO stage 3* | 0 (0.0) | 0 (0.0) | 0 (0.0) |  |
| Stroke/delirium | 19 (11.6) | 10 (8.5) | 9 (19.6) | 0.046 |
| Atrial fibrillation | 45 (27.4) | 32 (27.1) | 13 (28.3) | 0.883 |
| Perioperative MI | 16 (9.8) | 12 (10.2) | 4 (8.7) | 0.775 |
| IABP support | 5 (3.0) | 5 (4.2) | 0 (0.0) | 0.323 |
| ECMO support | 0 (0.0) | 0 (0.0) | 0 (0.0) |  |
| Pneumonia | 13 (7.9) | 9 (7.6) | 4 (8.7) | 0.758 |
| Peak troponin-I (ng/mL) |  | 2.2 (0.1–71.5) | 2.3 (0.1–58.1) | 0.608 |
| Ventilator support (hrs) |  | 17.2 (3.0–128.0) | 19.3 (6.0–101.0) | 0.041 |
| ICU stay (days) |  | 1.7 (0.6–14.8) | 1.9 (0.8–8.1) | 0.473 |
| Hospital stay (days) |  | 9 (5–144) | 9 (6–53) | 0.176 |
| 30-day mortality | 0 (0.0) | 0 (0.0) | 0 (0.0) |  |
| In-hospital mortality | 2 (1.2) | 1 (0.8) | 1 (2.2) | 0.484 |
| Late mortality | 9 (5.5) | 6 (5.1) | 3 (6.5) | 0.712 |

Values are n (%) or median (range). High: PWV ≥ 19 m/s, Low: PWV < 19 m/s. baPWV, brachial-ankle pulse wave velocity; ECMO, extracorporeal membrane oxygenation; IABP, intraaortic balloon pump; ICU, intensive care unit; KDIGO, Kidney Disease: Improving Global Outcomes; MI, myocardial infarction.

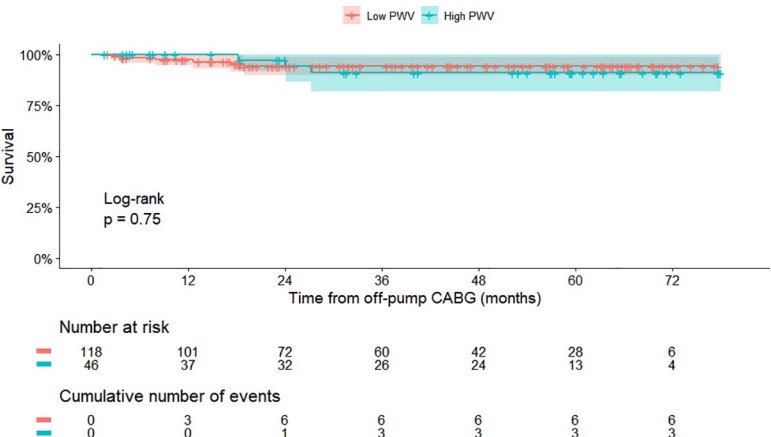

**Fig 5. Survival comparison between the patients with low and high pulse wave velocities.** CABG, coronary artery bypass grafting; PWV, pulse wave velocity.

complication) was higher in the high PWV group (19.6%, n = 9 vs. 8.5%, n = 10, $p$ = 0.046). The patients with high PWVs required longer durations of mechanical ventilatory support (19.3 hours [6.0–101.0 hours] vs. 17.2 hours [3.0–128.0 hours], $p$ = 0.041). There were no 30-day mortality in either group. However, one patient in each group died before discharge from the causes unrelated to the scope of our study, the hypovolemic shock with hemothorax and the pneumonia in a patient with colon cancer. The late all-cause mortality rates during the median follow-up of 39.2 months (range, 1.6–78.0 months) were not statistically different between the respective groups (6.5%, n = 3 vs. 5.1%, n = 6).

In terms of mid-term survival, the high PWV group had a 90.9% 5-year survival (95% CI, 81.7% - 100%) rate, whereas the low PWV group had a 94.2% 5-year survival (95% CI, 89.8% - 98.9%) rate, but this difference was not statistically significant ($p$ = 0.75) (Fig 5).

## Discussion

We demonstrated that postoperative AKI following off-pump CABG manifested mostly with mild features and was independently predictable by the non-invasive and simple measurement of baPWV. Moreover, baPWV predicted AKI better than preoperative serum creatinine concentration. For a secondary endpoint, high baPWVs ($\geq$ 19 m/s) were associated with the composite neurologic outcome and the duration of postoperative mechanical ventilatory support, but not with in-hospital mortality and mid-term survival.

The importance of renal insufficiency in association with cardiovascular mortality risk cannot be over-emphasized, both in surgical and non-surgical patients. Even minor reductions in GFR may lead to higher cardiovascular mortality rates [7] and even small increases in creatinine levels after CABG have been reported to raise the long-term risk of end-stage renal disease almost 3-fold [8]. Arterial stiffness is one of the possible mechanisms connecting renal insufficiency to cardiovascular events [9,10]. In this study, when the analysis was limited only to those patients with normal or mildly impaired preoperative renal function (eGFR $\geq$ 50 mLmin1.73m$^2$), there was a negative correlation between baPWV and postoperative eGFR, as depicted in Fig 2. From this association, it can be speculated that postoperative AKI associated with elevated PWV may affect long-term cardiovascular mortality. Therefore, patients with high PWV-associated AKI should have more thorough follow up.

To the best of our knowledge, our study is one of only two that directly investigated the association between post-CABG AKI and PWV [3,5]. The other study was a prospective

observational study with a similar design, and their findings were concordant with ours. The other group included 137 patients who underwent isolated CABG and carotid-femoral PWV assessment. Likewise, AKI was defined according to KDIGO practice guidelines. Their entry explanatory variables for multivariable analysis were eGFR, PWV, age, and sex; PWV and age were the final independent predictors. The odds ratio (OR) of developing AKI was 1.54, with every unit (m/s) increase in PWV aligning with our results (OR = 1.34; 95% CI, 1.17–1.58; $p < 0.001$). Further analysis regarding the association of PWV with postoperative outcomes was not described [3]. Limitations of that study included the fact that the authors did not elucidate whether the CABG procedures were off-pump, on-pump, or both, which adds the possibility of significant bias caused by the use of cardiopulmonary bypass.

Our observation that baPWV is associated with AKI independently of preoperative serum creatinine concentration or eGFR implies that arterial stiffness indicated by baPWV may affect postoperative renal function via its own mechanism, irrespective of baseline renal function. Although the mechanism is unknown, some inferences may be drawn. Patients with more severe arterial stiffness theoretically have higher pulse pressures. Because elevated pulse pressure can induce increases in afferent arteriolar tone and decreases in effective renal plasma flow [11], those patients are more likely to be afflicted with AKI following CABG, as fluctuating blood pressure can frequently aggravate renal perfusion. Another theory is related to development of type 2 and 4 cardiorenal syndrome. With increased arterial stiffness, reflected waves that assist with diastolic coronary artery filling return to the coronary artery os prematurely during late systole and compromise coronary blood flow under decreased diastolic blood pressure [12]. This can aggravate the relaxation disturbance associated with left ventricular (LV) hypertrophy and central remodeling [13,14]. The resultant elevated LV filling pressure may subsequently contribute to elevations in intra-abdominal venous pressure and a substantial reduction of renal blood flow and GFR [15].

The composite neurologic outcome composed of stroke and delirium was found to be associated with high baPWV, but there was no significant association observed between stroke and delirium separately. There are a lack of data testing the hypothesis that PWV is associated with the post-cardiac-surgery development of stroke or delirium. A recent study, examining post-aortic valve replacement neurocognitive dysfunction, demonstrated that patients with higher carotid-femoral PWV exhibited poorer performance in delayed memory, visual attention, response, and problem-solving tests [16]. Other reports have demonstrated the correlation between pulse pressure and stroke [17,18]. One study demonstrated an 11% increase of stroke risk for every 10 mmHg increase in pulse pressure [19]. Because higher PWVs can elicit elevated pulse pressures, it is speculated that patients with high PWVs may be more likely to have a stroke following CABG, and this may have manifested among the patients in our study. Additional studies with larger numbers of surgical patients are needed to clarify the strength and causality pattern of this relationship.

## Study limitations

There are some limitations that must be taken into consideration. Firstly, preoperative assessment of PWV was not routinely performed, especially in early period of the interval under study and in many urgent CABG cases. Thus, some study subjects who may have met the inclusion criteria were missed, contributing to selection bias. Secondly, most patients (84.8%) had near normal baseline kidney function (CKD stage 1 and 2); therefore, the predictive value of baPWV demonstrated by ROC curve analysis cannot be generalized to patients with more severe renal impairment. Thirdly, antihypertensive medications, such as vasodilators that the patients were taking before surgery, might have influenced the baPWV measurements, but

they were not controlled. Fourthly, we did not collect the complete data regarding periopera-tive hemodynamics and transfusion volume, which must have affected postoperative renal function. Lastly, we expect there would be a long-term correlation between renal function and baPWV in this cohort, but we could not investigate that because the related data were not available.

## Conclusions

BaPWV was an independent predictor of postoperative AKI following off-pump CABG, and high baPWV may affect the incidence of stroke and delirium (composite neurolgic outcome). These results need to be further investigated in studies including a much wider range of car-diac surgery patients.

## Supporting information

**S1 Table. Baseline clinical characteristics of low baPWV and high baPWV groups.**
(DOCX)

## Author Contributions

**Conceptualization:** Jae-Sung Choi, Jeong Sang Lee.

**Data curation:** Jae-Sung Choi, Se Jin Oh, Yong Won Sung, Hyeon Jong Moon, Jeong Sang Lee.

**Formal analysis:** Jae-Sung Choi, Jeong Sang Lee.

**Investigation:** Jae-Sung Choi, Jeong Sang Lee.

**Methodology:** Jae-Sung Choi, Jeong Sang Lee.

**Project administration:** Jae-Sung Choi, Jeong Sang Lee.

**Resources:** Jae-Sung Choi.

**Supervision:** Jae-Sung Choi, Jeong Sang Lee.

**Validation:** Jae-Sung Choi, Se Jin Oh, Yong Won Sung, Hyeon Jong Moon, Jeong Sang Lee.

**Visualization:** Jae-Sung Choi.

**Writing – original draft:** Jae-Sung Choi.

**Writing – review & editing:** Jae-Sung Choi.

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
