## [Decision Letter · Decision Letter 0]

20 Feb 2020

PONE-D-19-33416

Pulse Wave Velocity is a New Predictor of Acute Kidney Injury Development after Off-Pump Coronary Artery Bypass Grafting

PLOS ONE

Dear Dr Lee,

Thank you for submitting your manuscript to PLOS ONE. After careful consideration, we feel that it has merit but does not fully meet PLOS ONE’s publication criteria as it currently stands. Therefore, we invite you to submit a revised version of the manuscript that addresses the points raised during the review process.

In particular, as pointed out by Reviewer #2, it's not clear why you measured the baPWV before surgery in such a big cohort of patients. It seems to be a prospective study... reported as retrospective. Please clarify.

Moreover, I totally agree with Reviewer#3. I expect patients with a positive baPWV will have several additional risk factors for AKI. The Authors should clearly explain what, in their opinion, does baPWV add to current methods to predict the risk of AKI.

We would appreciate receiving your revised manuscript by 20/3/2020. To enhance the reproducibility of your results, we recommend that if applicable you deposit your laboratory protocols in protocols.io, where a protocol can be assigned its own identifier (DOI) such that it can be cited independently in the future. For instructions see: http://journals.plos.org/plosone/s/submission-guidelines#loc-laboratory-protocols

We look forward to receiving your revised manuscript.

Kind regards,

Laura Pasin

Academic Editor

PLOS ONE

Journal Requirements:

2. In ethics statement in the manuscript and in the online submission form, please provide additional information about the patient records/samples used in your retrospective study. Specifically, please ensure that you state whether the IRB or ethics committee waived the requirement for informed consent (rather than the authors themselves).

3. We note that you have reported significance probabilities of 0 in places. Since p=0 is not strictly possible, please correct this to a more appropriate limit, eg 'p<0.0001'.

5. Your ethics statement must appear in the Methods section of your manuscript. If your ethics statement is written in any section besides the Methods, please move it to the Methods section and delete it from any other section. Please also ensure that your ethics statement is included in your manuscript, as the ethics section of your online submission will not be published alongside your manuscript.

Reviewers' comments:

Reviewer's Responses to Questions

**Comments to the Author**

1. Is the manuscript technically sound, and do the data support the conclusions?

Reviewer #1: Yes

Reviewer #2: Yes

Reviewer #3: Partly

2. Has the statistical analysis been performed appropriately and rigorously? 

Reviewer #1: I Don't Know

Reviewer #2: Yes

Reviewer #3: Yes

3. Have the authors made all data underlying the findings in their manuscript fully available?

Reviewer #1: Yes

Reviewer #2: Yes

Reviewer #3: No

4. Is the manuscript presented in an intelligible fashion and written in standard English?

Reviewer #1: Yes

Reviewer #2: No

Reviewer #3: Yes

5. Review Comments to the Author

Reviewer #1: The proposed work is very interesting. It is a significant contribution to the possibility of predicting acute kidney failure in the cardiac surgical patient's setting.

However there are some considerations that reserve attention.

First of all, it is not specified when the study was done, in the preoperative, postoperative period? It’s unclear why the conditions change greatly and are influenced by various factors. It would be desirable to define exactly when it was done and whether patients had drug infusions. Not only does preoperative therapy affect the measurements.

Moreover, the technique is not described at all. It is important to understand how arterial stiffness is calculated.

Again, Table 1 should be done separately by separating baseline data from operative data.

The abbreviations must be specified in the tables.

Finally, figures 1 and 2 need to be redone to be clearer.

Reviewer #2: The authors present with a study exploring the role of baPWV as a risk factor predictor for AKI after off-pump CABG surgery.

Being a non-invasive and easily reproducible technique, the baPWV can be an interesting screening tool for AKI risk assessment purpose.

Some suggestions:

- given the retrospective nature of the study, the authors should clarify why they measured the baPWV before surgery in such a big cohort of patients

- patient selection, described in the Study population section, should be reported as a flowchart

- Table 1: creatinine, GFR and euroscore II means have high SD values and seem not to be normally distributed. Did the authors test continuous variables for normality? They did not report the adoption of any specific test for this issue.

- more data should be reported as regard the surgical procedure (lenght of surgery, type of CABG)

- baPWV AUC resulted to be superior among the creatinine AUC but the difference is not really high. The authors should give to the readers more reasons to use one more diagnostic tool besides the creatinine measure. Maybe it could be useful to report also the negative and positive predicting values of baPWV.

- in the limitations section should be underlined also the retrospective design of the research

- the text needs a linguistic revision

Reviewer #3: In their manuscript, dr. Lee and colleagues present results of a retrospective observational study investigating the role of baPWV as a predictor of postoperative AKI in patients undergoing off-pump CABG.

The rationale for the study is sound, and the topic is potentially relevant as AKI is a common complication following cardiac surgery.

My major comment concerning this manuscript is that I expect patients with a positive baPWV (i.e. patients with diffuse atherosclerotic disease) will also have several additional risk factors for AKI. The Authors should better explain what, in their opinion, does baPWV add to current methods to estimate the risk of AKI

Furthermore, I have other comments which I hope will help the Authors to improve their manuscript:

1. Please present the same data of Table 1 divided by high baPWV and low baPWV patients. This table can be presented in a supplementary appendix

2. AKI is common after cardiac surgery and has a clear prognostic impact. Therefore, I agree with the Author in chosing AKI stage 1 as primary outcome. However, I would like to see also the correlation between baPWV and development of stage 3 AKI, as well as correlation with need for postoperative renal-replacement therapy, if there are sufficient data.

In any case, please present in table 2 data on AKI divided by stage, and need for RRT

3. Please specify the follow-up and clearly define "late" death"

4. Do the Authors have data on long-term ("late") kidney function? Would it be possible to analyze the correlation between baPWV and long-term renal outcome?

5. Please provide data on sample size calculation

6. The association between baPWV and stroke/delirium is borderline. Please acknowlegde this in the limitation. Of note, this is an interesting finding and a separate study on this would be interesting

7. What was the decision to measure baseline baPWV made on? Is it possbile that very low-risk patients did not have baPWV measured and were therefore excluded from the study? How would this impact on study results, in the Authors' opinion?

8. ICU stay and lenght of hospital stay are relatively long compared to modern cardiac surgery. Please explain

9. Please provide a definition for postoperative pneumonia

10. For postoperative MI, the definition is not clear. In particular, the role of troponin is not clear. Could a MI be diagnosed only with troponin, or did it require an association between troponin and CK-MB, or troponin and ECG changes?

11. There are some laguage mistakes across the manuscript. Please have the manuscript reviewed by a native english speaker with expertise in biomedical scientific writing

6. PLOS authors have the option to publish the peer review history of their article (what does this mean?). If published, this will include your full peer review and any attached files.

Reviewer #1: Yes: Pittarello Demetrio

Reviewer #2: No

Reviewer #3: Yes: Alessandro Belletti

---

## [Author Response · Author response to Decision Letter 0]

23 Mar 2020

Dear Professor Laura Pasin and the reviewers,

We, the authors of this study, have read and discussed the comments that you and the reviewers made. We think that the comments helped improve the quality of this work. For the matter of our study design pointed out by you and Reviewer #2, in fact, this study has both prospective and retrospective elements. It is true that the PWV data had been prospectively collected before the operations. On the other hand, some of these data had been already collected for some other study purposes in the cardiology department before surgical consultations were needed, and all of the data existed at the time our protocol was submitted to the IRB for initial approval. In other words, we retrospectively analysed the prospectively collected data. Considering the opinion of the editor and reviewer, we decided to remove the phrase, “In our retrospective study”, from the method section of the Abstract and to replace it with “… baPWV data that were prospectively collected from 164 patients …”. The fact that retrospective analysis was performed was expressed indirectly in the Methods section of the main body: “informed consent was waived because this retrospective review of medical records could not adversely affect the rights or welfare of the subjects.”

You and Reviewer #3 requested a clear explanation for what baPWV adds to current methods to predict the risk of AKI. We think the request was based on the rational expectation that a high baPWV would be associated with some additional risk factors contributing to AKI. Although we scoured the literature, we did not find a better explanation than what we had already described in the Discussion section. In a nutshell, the additional risk of AKI associated with high baPWV, for now, is associated with an increased afferent arteriolar tone and an elevated left ventricular filling pressure. The speculated mechanisms, in our opinion, are clearly described with the references in the Discussion section (page 15, line # 288 – page 16, line # 302). 

Additionally, we should mention that the mechanical ventilatory support durations were longer among patients with high baPWVs. Because these duration data were not normally distributed, we changed the statistical methods to incorporate the use the Mann–Whiney test instead of t-tests, which led to statistical significance. This fact is now described in the Results section and in the abstract. This is the only statistical consideration that has changed after the revision.

For Reviewer #1’s comments, firstly, our primary endpoint was investigating whether or not postoperative development of AKI was associated with preoperative baPWV and then how much and how independently they were associated. We did specify when this study was conducted with the use of “preoperatively” in the first sentence of the subsection, “Measurement of brachial-ankle pluse wave velocity”: “baPWV measurements were preoperatively performed using a volume-plethysmographic apparatus... .“ Secondly, it is right that PWV can be influenced by various factors like inotropic drugs and patients’ hemodynamic status postoperatively, but the PWV measurements that were dealt with in this study were all done preoperatively. However, even preoperative PWV might be influenced by what kind of medications patients are taking. So, we modified the sentence in the subsection, “Study limitations”: “antihypertensive medications, such as vasodilators that the patients were taking before surgery, might have influenced the baPWV measurements, but they were not controlled.” Thirdly, the baPWV calculation method was described in detail in the subsection, “Measurement of brachial-ankle pulse wave velocity”: “To calculate the baPWV, the virtual length of the arterial path between the heart and the brachial artery (Lhb) and the length of the path between the heart and the posterior tibial artery (Lha) should be derived from the subject’s height using the appropriate formulas. Then, the time difference between the beginning points of systolic increase in brachial and ankle pressure waves (Tba) should be measured. Finally, baPWV was calculated by the path length difference devided by the time difference (baPWV = (Lha – Lhb)/Tba).” Fourthly, we separated the operation data (Table 2) from the baseline data (Table 1), and, with that, some more important data were added to the respective Tables. Lastly, Figures 1 and 2 have been modified for clarity.

For Reviewer #2’s comments, firstly, the changes in the text we made regarding the study design have already described in the letter to the editor. We can say that this work involved a retrospective analysis of prospectively collected data. Secondly, we have added a patient selection flowchart in Fig 1. Thirdly, as Reviewer #2 pointed out, creatinine level, GFR, and EuroSCORE II were not normally distributed. So, these data have been presented as median (range) instead of mean values, and the comparison was performed by a nonparametric method, the Mann-Whitney U test. These results have been presented in Table 1. Of course, the statistical results have not been changed. The relevant text has also been corrected in accordance with these results in the statistical subsection and in the subsection, “Association of baseline characteristics with postoperative AKI” We did the same thing with some other data that were not normally distributed. With this, more surgical data about the off-pump CABG were added and reported in a separate table (Table 2). Besides, the surgical technique used during the study period was introduced in the subsection, Surgical techniques. Fourthly, we provided the positive and negative predictive values of high baPWV for AKI and compared these with those of creatinine. This is described in the subsection, “Independent predictors of postoperative AKI”: “Moreover, using preoperative baPWV and serum creatinine cut-off values of 18.2 m/s and 1.0 mg/dL, respectively, from the respective ROC curves, the positive and negative predictive values for baPWV are 43.5% and 91.5%, respectively, compared with 32.7% and 88.4%, for creatinine. Therefore, baPWV is likely to be more useful in terms of predicting postoperative AKI.” Lastly, this study had some elements of retrospective design with their own limitations. We think that these limitations have already been mentioned. The whole text was thoroughly reviewed by a native English speaker with expertise in this field.

For Reviewer #3’s comments, first of all, we appreciate that the reviewer positively touched on every detail to improve this manuscript. For comment #1, we presented additional baseline data of the patients with high and low baPWV in a supplementary appendix, for reference. This supplementary table was specified in the subsection, “Impact of high baPWV on postoperative outcomes”: “…despite the fact that preoperative creatinine levels of the high PWV group was not statistically different from those of the low PWV group, and preoperative median eGFR value of the low PWV group was 75.7 mL/min/1.73 m2 which was not bad. For further details about comparison of baseline characteristics between the high and the low baPWV groups, please see supplementary material (S1 Table).” For comment #2, we also wanted to see the correlation between the high baPWV and the development of more advanced AKI, but there were no instances of stage 3 AKI or renal replacement therapy in this patient cohort. Even the incidence of stage 2 AKI was insufficient to perform a valid analysis. Anyway, postoperative AKI data were stratified by stage and presented in the Table 4. For comment #3, we specified follow-up details in the subsection, “Patient selection”: “The median follow-up duration was 39.2 months (range, 1.6 - 78.0 months). There was no loss to follow-up among the included patients.” Late mortality was as follows: “Late mortality was defined as any-cause mortality after POD 30” in the subsection, “Renal function and postoperative complications.” For comments #4 & #5, like the results of some studies dealing with the impact of postoperative AKI on long-term renal function, we expect there would be a long-term correlation between renal function and baPWV even in this cohort. For example, Ryden et al.* found that even a small increase in serum creatinine after CABG was associated with a 3-fold increased risk of ESRD development in the long-term. But we think that this is beyond the scope of our study focusing on postoperative AKI. (*Rydén L, Sartipy U, Evans M, Holzmann MJ. Acute Kidney Injury After Coronary Artery Bypass Grafting and Long-Term Risk of End-Stage Renal Disease. Circulation. 2014;130(23):2005-11). For comment #6, we think that many things about the borderline inter-group difference of the composite neurologic outcome composed of stroke and delirium, have already been covered in the Discussion section, including an acknowledgement of the borderline significance, a brief introduction of recent relevant publications, and the necessity for additional studies to elucidate this correlation. Therefore, do you really think we still need to place the same acknowledgement again in the limitations subsection? For comment #7, concerning this selection bias, we never had PWV measurements skipped only because patient risk was very low. Some of the measurements were not performed because it was just not a routine preoperative evaluation during the early period, some measurements were refused by patients due to the fact that the cost was not being covered by insurance, and some were skipped in urgent or emergency contexts. This is briefly mentioned in the limitations subsection. This is one of the typical limitations of retrospective studies that do not include prospective controls. For comment #8, the relatively long median durations of ICU and hospital admissions in our study were mainly due to the inclusion of several outliers. Because these duration data were not normally distributed, we compared the median values rather than mean values using the Mann-Whitney non-parametric test, showing a little shorter duration (ICU stay and hospital stay, 1.7 and 9 days, respectively, in low baPWV group vs. 1.9 and 9 days in high baPWV group, p = ns). This has been presented in Table 4. However, you may still think these durations looks a bit long. We think this was due to several factors, like manpower shortages, patients’ extreme reluctance to be discharged early, rare pressure from the management of our city government hospital. For example, the vast majority of the patients undergoing cardiac surgery do not move to a general ward on the day of surgery, regardless of how fast the extubation is done and how good the patient’s condition is. For comment #9, postoperative pneumonia is now defined as follows: “Postoperative pneumonia was defined as a lower respiratory tract infection with accompanying consolidation detected on chest x-ray,” in the subsection, “Renal function and postoperative complications.” For comment #10, PMI (perioperative myocardial infarciton) was not diagnosed only with troponin level, but with considering 3 factors: CK-MB, troponin I, and EKG findings. We have already described this in the subsection, “Renal function and postoperative complications”: “Perioperative myocardial infarction was defined as creatine kinase (CK-MB) concentration >40 ng/mL, new Q waves noted on electrocardiogram, or peak troponin I levels >15 ng/mL at 12 hours after operation.” For comment #11, this entire manuscript was thoroughly reviewed by a native English speaker with expertise in this field.

Sincerely Yours,

Jae-Sung Choi, MD, PhD, 

Clinical professor, SNU-SMG Boramae Medical Center, Seoul, Korea

Jeong Sang Lee, MD, PhD

Professor, Dept. of Thoracic & Cardiovascular Surgery, Seoul National University College of 

Medicine, SNU-SMG Boramae Medical Center, Seoul, Korea

---

## [Decision Letter · Decision Letter 1]

6 Apr 2020

PONE-D-19-33416R1

Pulse Wave Velocity is a New Predictor of Acute Kidney Injury Development after Off-Pump Coronary Artery Bypass Grafting

PLOS ONE

Dear MD,PhD Lee,

Thank you for submitting your manuscript to PLOS ONE. After careful consideration, we feel that it has merit but does not fully meet PLOS ONE’s publication criteria as it currently stands. Therefore, we invite you to submit a revised version of the manuscript that addresses the points raised during the review process.

Dear Dr. Jeong Sang Lee,

you have addressed most of Reviewer's comments. There are only few minor issues that need to be addressed. Please be clear and precise in you next revision. You did a very good job.

We would appreciate receiving your revised manuscript by May 21 2020 11:59PM. To enhance the reproducibility of your results, we recommend that if applicable you deposit your laboratory protocols in protocols.io, where a protocol can be assigned its own identifier (DOI) such that it can be cited independently in the future. For instructions see: http://journals.plos.org/plosone/s/submission-guidelines#loc-laboratory-protocols

We look forward to receiving your revised manuscript.

Kind regards,

Laura Pasin

Academic Editor

PLOS ONE

Reviewers' comments:

Reviewer's Responses to Questions

**Comments to the Author**

1. If the authors have adequately addressed your comments raised in a previous round of review and you feel that this manuscript is now acceptable for publication, you may indicate that here to bypass the “Comments to the Author” section, enter your conflict of interest statement in the “Confidential to Editor” section, and submit your "Accept" recommendation.

Reviewer #1: All comments have been addressed

Reviewer #2: All comments have been addressed

Reviewer #3: (No Response)

2. Is the manuscript technically sound, and do the data support the conclusions?

Reviewer #1: Yes

Reviewer #2: Yes

Reviewer #3: Yes

3. Has the statistical analysis been performed appropriately and rigorously? 

Reviewer #1: Yes

Reviewer #2: Yes

Reviewer #3: Yes

4. Have the authors made all data underlying the findings in their manuscript fully available?

Reviewer #1: Yes

Reviewer #2: No

Reviewer #3: Yes

5. Is the manuscript presented in an intelligible fashion and written in standard English?

Reviewer #1: Yes

Reviewer #2: Yes

Reviewer #3: Yes

6. Review Comments to the Author

Reviewer #1: The authors responded to the questions raised comprehensively. I therefore think that the proposed work is acceptable.

Reviewer #2: The authors addressed all reviewers' criticism. The manuscript reads well and the topic is clearly presented.

Reviewer #3: In this manuscript, dr. Lee and colleagues present a revised version of their work.

Overall, the Authors have adequately addressed most of my commments. I only have few minor comments:

1. The Authors have not addressed the issue of sample size calculation

2. Although no patient required RRT, please specify this overtly in the Results

3. The definition of MI remains ambiguous. Please specify that patients should have both ECG findings AND positive biomarkers

4. Please ackowledge that no data on long-term kidney function were available

7. PLOS authors have the option to publish the peer review history of their article (what does this mean?). If published, this will include your full peer review and any attached files.

Reviewer #1: Yes: Pittarello Demetrio

Reviewer #2: No

Reviewer #3: Yes: Alessandro Belletti

---

## [Author Response · Author response to Decision Letter 1]

10 Apr 2020

Dear Professor Laura Pasin,

We, the authors of this study, have read and discussed the new comments from the reviewer #3 and we made a minor revision. 

For the issue of sample size calculation, because this study was neither a typical prospective nor a randomized controlled trial, we did not predetermine a sample size. As the reviewer #3 knows well, unlike prospective randomized controlled studies, retrospective review studies usually use statistical power rather than the calculation of sample sizes, thus, we performed the post hoc power analysis. The power (1-β) that we calculated was 99.3% for the statistical comparison of AKI incidences between the low baPWV group and high baPWV group, and 84.0% for the comparison of mean baPWV between the AKI group and no-AKI group. By the way, although the powers are fairly acceptable, we don’t think that reporting this fact is necessary because the observed powers (or post-hoc) are directly related to the p-values that were already demonstrated as < 0.001 in our study. In short, we don’t need to distinguish between true negative and false negative with a post-hoc power for our positive study results. 

For the asking to specify that no postoperative RRT event was observed, we presented this fact in the subsection, “ Impact of high baPWV on postoperative outcomes”: “There was no patient who required renal replacement therapy in patients with postoperative AKI though.” 

For the ambiguous definition of PMI, we redescribed it as, “Perioperative myocardial infarction was defined based upon an elevation of biomarkers (either creatine kinase (CK-MB) concentration >40 ng/mL or peak troponin I levels >15 ng/mL at 12 hours after operation) and the presence of new pathological Q waves or left bundle branch block.” 

For the lack of data on long-term relationship between renal function and baPWV, we add this fact to the limitation subsection as, “Lastly, we expect there would be a long-term correlation between renal function and baPWV in this cohort, but we could not investigate that because the related data were not available.” 

Sincerely Yours,

Jae-Sung Choi, MD, PhD, 

Clinical professor, SNU-SMG Boramae Medical Center, Seoul, Korea

Jeong Sang Lee, MD, PhD

Professor, Dept. of Thoracic & Cardiovascular Surgery, Seoul National University College of 

Medicine, SNU-SMG Boramae Medical Center, Seoul, Korea

---

## [Editor Report · Decision Letter 2]

14 Apr 2020

Pulse Wave Velocity is a New Predictor of Acute Kidney Injury Development after Off-Pump Coronary Artery Bypass Grafting

PONE-D-19-33416R2

Dear Dr. Jeong Sang Lee,

We are pleased to inform you that your manuscript has been judged scientifically suitable for publication and will be formally accepted for publication once it complies with all outstanding technical requirements.

With kind regards,

Laura Pasin

Academic Editor

PLOS ONE
---

## [Editor Report · Acceptance letter]

17 Apr 2020

PONE-D-19-33416R2 

Pulse Wave Velocity is a New Predictor of Acute Kidney Injury Development after Off-Pump Coronary Artery Bypass Grafting 

Dear Dr. Lee:

I am pleased to inform you that your manuscript has been deemed suitable for publication in PLOS ONE. Congratulations! Your manuscript is now with our production department. 

With kind regards,

on behalf of

Dr. Laura Pasin 

Academic Editor

PLOS ONE